# New Insights into the Catalytic Activity of Second Generation Hoveyda–Grubbs Complexes Having Phenyl Substituents on the Backbone

**Assunta D'Amato** [1] , **Annaluisa Mariconda** [2,*] **and Pasquale Longo** [1]

1   Department of Chemistry and Biology, University of Salerno, Via Giovanni Paolo II, 132, 84084 Fisciano, Italy; asdamato@unisa.it (A.D.); plongo@unisa.it (P.L.)
2   Department of Science, University of Basilicata, Viale dell'Ateneo Lucano, 10, 85100 Potenza, Italy
*   Correspondence: annaluisa.mariconda@unibas.it

**Abstract:** One of the most effective synthetic pathways to produce unsaturated compounds and polymers, meant for both industrial and pharmaceutical applications, is olefin metathesis. These useful reactions are commonly promoted by ruthenium-based precatalysts, namely the second-generation Grubbs' catalyst (GII) and complexes bearing a styrenyl ether ligand, referred to as the second-generation Hoveyda–Grubbs' catalyst (HGII). By altering the steric and electronic characteristics of substituents on the backbone and/or on the nitrogen atoms of the NHC ligand, it is possible to increase the reactivity and stability of second-generation ruthenium catalysts. The synthesis of an HG type II complex bearing two *anti*-phenyl backbone substituents (*anti*-HGII$_{Ph-Mes}$) with mesityl N-substituents is reported. The catalytic performances of the new complex were investigated in standard ring-closing metathesis (RCM) and ring-opening metathesis polymerization (ROMP) and compared to those of the analogue complex *syn*-HGII$_{Ph-Mes}$ and to the classic HGII complex. A thorough analysis of the temperature dependence of the performances, along with a detailed comparison with the commercially available HGII, is conducted. The HGII$_{Ph-Mes}$ complexes are more thermally stable than their parent HGII, as shown by the fact that their activity in the ROMP of 5-ethylidene-2-norbornene does not alter when the polymerization is carried out at room temperature after the complexes have been held at 180 °C for two hours, making them particularly interesting for materials applications.

**Keywords:** NHC-Ru(II); ligand design; RCM; ROMP

## 1. Introduction

Olefin metathesis is an effective and versatile synthetic method for the preparation of different compounds, including polymers, natural products, and biologically active substances [1–6]. It has a very wide range of industrial applications, such as biochemicals [7–9], pharmaceuticals [10–12], cosmetics [13–15], agrochemicals [16–21], and advanced materials [22–26]. Olefin metathesis reactions [1,2] are classified into three categories: cross metathesis (CM), ring-closing metathesis (RCM), and ring-opening metathesis (ROM). The accepted reaction mechanism, characterized by the formation of a metallacyclobutane intermediate, was first proposed by Chauvin [27].

Among the various metal-based precatalysts used in these reactions, the well-defined ruthenium–carbene complexes, known as Grubbs' catalysts, display exceptional functional group tolerance and have been extensively studied in olefin metathesis. The development of very active systems, also useful for an effective metathesis of electron-deficient substrates, involved ruthenium compounds with 1,3-dimesityl-4,5-dihydroimidazol-2-ylidene as a carbene ligand (second-generation Grubbs' catalyst—GII) and complexes bearing a styrenyl ether ligand (second-generation Hoveyda-Grubbs' catalyst—HGII) [28,29] (Figure 1).

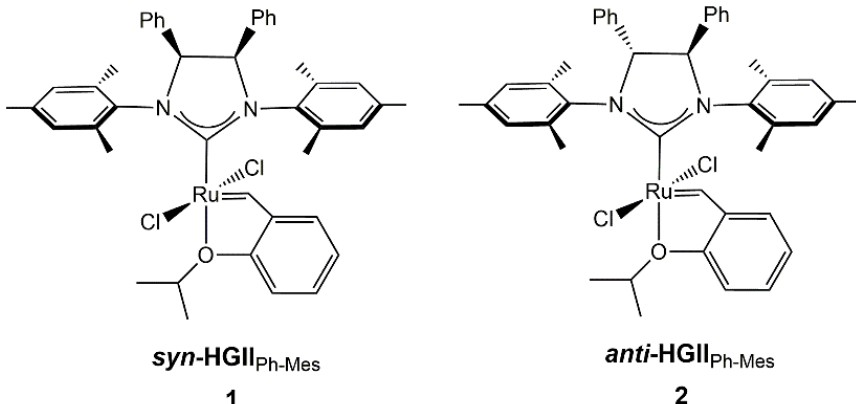

**Figure 1.** Second generation Grubbs' (**GII**) and Hoveyda–Grubbs' (**HGII**) catalysts.

The reactivity and stability of second-generation ruthenium catalysts can be improved by modifying the steric and electronic properties of substituents on the backbone and/or on the nitrogen atoms of the NHC ligand [30,31].

Several studies have shown that decreasing the size of the aryl groups on the NHC nitrogen atom is advantageous for RCM and CM of hindered substrates [32,33], while unsymmetrical NHC ruthenium(II) catalysts display beneficial properties, including strong thermodynamic stability, chemical latency, and exceptional selectivity in several specialized metathesis processes [34,35].

According to several research publications, the presence of backbone substituents on NHC ligands results in more stable complexes since they hamper the rotation of the substituents on the nitrogen atoms, thus limiting the decomposition of complexes by C–H activation [36–41]. Furthermore, the electronic effects of the substituents of the NHC backbone enhance the σ donor ability of carbene carbon [34].

A catalyst with these features has been used for the development of smart composite materials capable of self-repair in aeronautical structures. In fact, some of us synthesized and characterized a stable initiator (**1**) for ROMP reactions, suitably designed to be embedded in structural resins for self-healing applications (Figure 2) [42].

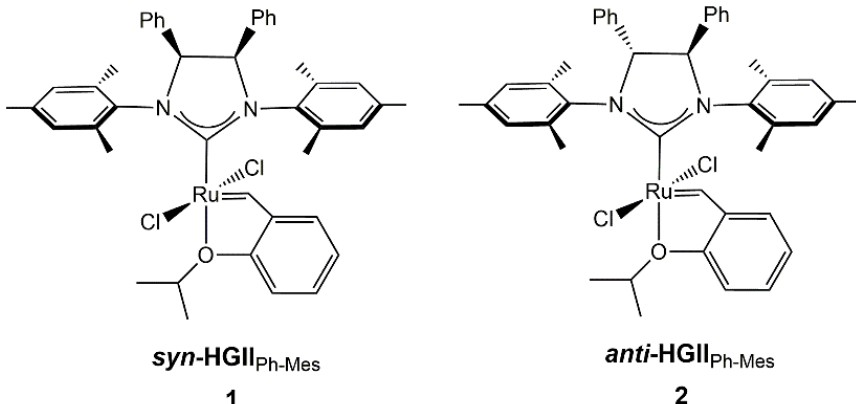

**Figure 2.** Hoveyda–Grubbs type catalysts.

Interesting results were obtained by solubilizing *syn*-HGII_{Ph-mes} in epoxy matrix structural resins, hardened by curing cycles up to 180 °C. Despite the difficult environment represented by this thermosetting material and the high temperatures to which the catalyst is exposed during the curing cycles, it was found active for the ROMP of 5-ethylidene-2-norbornene (ENB), which is the monomer used for the self-healing of these composite materials in the aviation sector [43]. The same tests performed in the presence of *anti*-HGII_{Ph-Mes} (**2**) (Figure 2) give different results. In fact, the latter catalyst is active in this hostile environment, but only if cured at temperatures up to 90 °C [44].

In this paper, we report the details of the synthesis and characterization of *anti*-HGII_{Ph-Mes} (**2**) and the comparison of *syn*- and *anti*-HGII_{Ph-Mes} activities with respect to the

commercial complex HGII (Figure 1) in the ring-closing metathesis and in the ring-opening metathesis polymerizations.

## 2. Results and Discussion

### 2.1. Synthesis and Characterization of Anti-HGII$_{Ph\text{-}Mes}$ Complex

The synthesis of complex *anti*-HGII$_{Ph\text{-}Mes}$ (**2**) is described in Scheme 1. It was obtained using the same procedure developed in Ref. [38] to obtain **1** *syn*-HGII$_{Ph\text{-}Mes}$. The enantiomerically pure ruthenium complex is prepared in three steps by using the slightly modified procedure reported in Ref. [41]. 1*R*,2*R*-diphenylethylenediamine was coupled with 2-bromomesitylene by palladium-catalyzed reaction to give diarylated diamine A ((1*R*,2*R*)-*N*,*N*′-di(mesityl)-1,2-diphenylethylenediamine) as an orange dusty solid in 92% yield. The $^1$H and $^{13}$C NMR spectra of A are consistent with the structure reported in the literature. The doublet signals at $4.7_8$ and $4.0_1$ ppm in the $^1$H NMR spectrum are attributed to NCHPh, and the signals at 141.2 and 141.9 ppm in the $^{13}$C NMR spectrum are attributed to quaternary carbons of the mesityl and phenyl groups, respectively [41].

**Scheme 1.** Synthesis of **2** *anti*-HGII$_{Ph\text{-}Mes}$.

The resulting diamine is reacted with triethyl orthoformate and ammonium tetrafluoroborate to form the imidazolium tetrafluoroborate salt B. The cyclization is confirmed by the signals at $8.3_6$ ppm in the $^1$H- and at 158.6 ppm in the $^{13}$C NMR spectra, attributable to hydrogen and carbon, respectively, of the cationic *CH* methine group between the nitrogen atoms. The treatment of B with potassium-hexamethyldisilazide (KHMDS) generates the carbene, which was not isolated and was reacted with dichloro (*o*-isopropoxyphenylmethylene) (tricyclohexylphosphine)ruthenium (II) (HGI) to give an enantiopure complex *anti*-HGII$_{Ph\text{-}Mes}$ as a bright green solid (30% 38 yield). The reaction was monitored by $^1$H NMR following the disappearance of the methine proton of B because of the formation of the carbenic carbon and the shift of the benzylidenic proton of *anti*-HGII$_{Ph\text{-}Mes}$ respect to that of HGI. The complex, purified by flash chromatography, was stable in the solid state. The $^1$H NMR analysis of the complex shows the characterization signal of the benzylidene proton at $16.3_5$ ppm, and the $^{13}$C NMR spectrum displays signals at 214.5 and 298.3 ppm due to the carbene carbons of the *N*-heterocyclic and benzylidene, respectively.

Mass Spectrometry (ESI-MS) confirmed the obtaining of the ruthenium complex, with the peak corresponding to the values of ESI$^+$MS $m/z$ of 742.90 Dalton in accordance with the mass of the fragment $[C_{43}H_{47}ClN_2ORu]^+$.

### 2.2. Metathesis Reactions: Activity Studies

#### 2.2.1. Ring Closing Metathesis Activity Studies

The catalytic activity in the olefin metathesis processes may be affected by the distinct phenyl dispositions (*syn* and *anti*) on the backbone. In order to assess the activities of the two precatalysts, *syn*-HGII$_{Ph\text{-}Mes}$ and *anti*-HGII$_{Ph\text{-}Mes}$, benchmark RCM and ROMP reactions were performed. Thus, we decided to evaluate the activities of the two complexes and compare them with those of the classic Hoveyda–Grubbs II complex. The ring-closing metathesis of diethyl-diallylmalonate C (Figure 3) and diethyl-allylmethallylmalonate **E** (Figure 4) was carried out in the presence of 0.1 mol% of ruthenium catalysts in $C_6D_6$ at 60 °C. The conversion of each olefin to a product was checked over time by $^1$H NMR

spectroscopy, and the corresponding kinetic profiles are shown in Figures 3 and 4. For RCM of C, all precatalysts showed similar catalytic activities: HGII reaches maximum conversion within 4 min, while for *syn-* and *anti-*HGII$_{Ph-Mes}$, the conversions are about 95% and 98% for the same time, respectively.

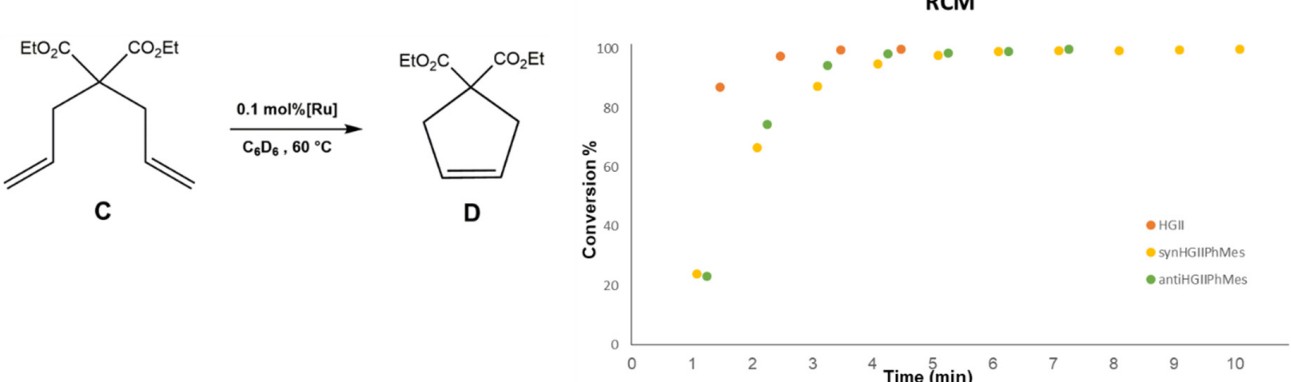

**Figure 3.** Kinetic data for the RCM of diethyl-diallylmalonate (C).

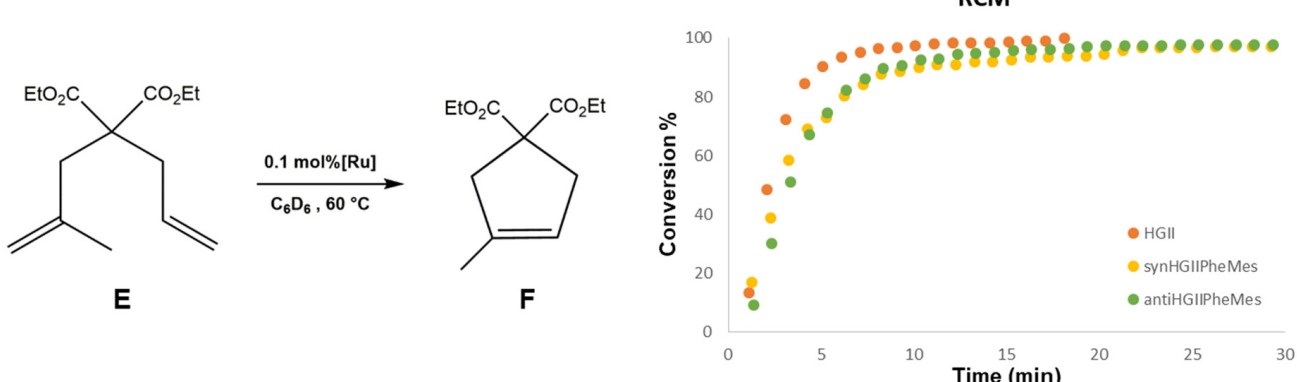

**Figure 4.** Kinetic data for the RCM of diethyl-allylmethallylmalonate (E).

For the RCM of E, precatalyst HGII was found to be slightly more active than *syn-* and *anti-*HGII$_{Ph-Mes}$: diethyl-allylmethallylmalonate total conversion was observed in 18 min, whereas *syn-* and *anti-*HGII$_{Ph-Mes}$ reached 97% and 98% conversion within 26 min, respectively. So, although highly efficient, precatalysts *syn-* and *anti-*HGII$_{Ph-Mes}$ are slightly less active than classical HGII for these reactions.

In the more sterically demanding RCM of diethyl-dimethylallylmalonate G, a 5% mol catalyst loading was employed in C$_6$D$_6$ at 60 °C (Table 1). There were marked differences in the reactivity among the three catalysts. After one hour, HGII converts to 20%, whereas precatalysts *anti-*HGII$_{Ph-Mes}$ and *syn-*HGII$_{Ph-Mes}$ convert to 11% and 4%, respectively. After 72 h, the activities of the three catalysts are only slightly different; in fact, HGII achieves a conversion of 63%, while *anti-*HGII$_{Ph-Mes}$ and *syn-*HGII$_{Ph-Mes}$ reach about 60 and 55%, respectively. As a result, whereas *syn-*HGII$_{Ph-Mes}$ and *anti-*HGII$_{Ph-Mes}$ exhibit lower initial activity than HGII, over a long period of time, they behave similarly.

In the literature, the observed trend is that sterically bulky catalysts [38,45,46] are significantly slower in the RCM of hindered olefins.

**Table 1.** RCM conversion of diethyl-dimethylallylmalonate (**G**).

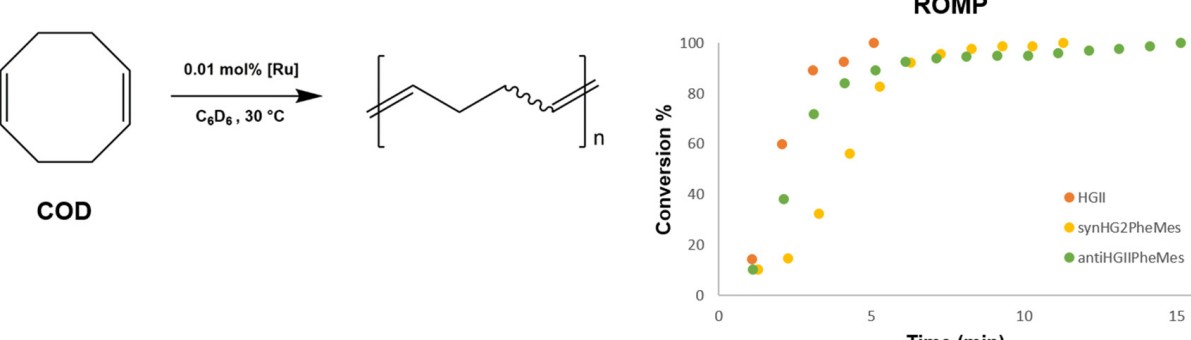

| Time (h) | Conversion % HGII | Conversion % *syn*-HGII$_{Ph-Mes}$ | Conversion % *anti*-HGII$_{Ph-Mes}$ |
|---|---|---|---|
| 1 | 20.0 [47] | 4.40 | 11.1 |
| 24 | 54.5 | 44.8 | 51.0 |
| 48 | 60.1 | 52.7 | 56.5 |
| 72 | 63.3 | 55.1 | 59.8 |
| 96 | 63.5 | 55.1 | 61.8 |

### 2.2.2. Ring-Opening Metathesis Polymerization Activity Studies

Successively, we tested the catalytic activity of *syn*- and *anti*-HGII$_{Ph-Mes}$ in the Ring Opening Metathesis Polymerization (ROMP) of 1,5-cyclooctadiene (COD), 2-norbornene (N), and 5-ethyliden-2-norbornene (ENB).

Figure 5 shows the ROMP of COD kinetic profiles in the presence of 0.01 mol% HGII, *syn*- and *anti*-HGII$_{Ph-Mes}$ ruthenium complexes to induce polymerization of COD by ring opening, providing the same polymer produced by 1,4-polyaddition of 1,3-butadiene. Both *syn*- and *anti*-HGII$_{Ph-Mes}$ are efficient catalysts for the polymerization of COD, reaching full monomer conversion within 11 and 15 min, respectively. The compound HGII gives the same result in 5 min. Thus, observing the kinetic profiles of these ruthenium-based pre-catalysts, we can affirm that *syn*- and *anti*-HGII$_{Ph-Mes}$ complexes are slightly less active than the complex lacking the substituents on the backbone. Furthermore, by observing the ROMP kinetic profile of COD with *syn*-HGII$_{Ph-Mes}$, the catalyst's need for an induction time is evident (see Figure 5).

**Figure 5.** Kinetic data for the ROMP of 1,5-cyclooctadiene (**COD**).

Probably, because in this compound, due to steric effects, the coordination of the incoming monomer could be more difficult than that in complex HGII and in *anti*-HGII$_{Ph-Mes}$, and maybe this could make the elimination of the ligand in *trans* to NHC harder. In other words, we can consider the coordination of the incoming monomer as a nucleophilic substitution of oxygen by the isopropoxide group by an associative or interchange mechanism.

It may be useful to recall that, as reported by Grisi et al. [48], the stability and reactivity of complexes with different substituents on the nitrogen atoms of NHC are strongly influenced by the *syn* or *anti*-configuration of the phenyl groups on the ligand backbone.

In fact, the complex with the phenyls in the *anti*-configuration is more stable and active than the homologue in the *syn* configuration. In the present work, only the stereochemical effects of the phenyls in positions 4 and 5 are evaluated because the nitrogen atoms have the same mesityl substituents. Our catalysts are highly efficient in both RCM catalysis of olefins and ROMP.

ROMPs of 2-norbornene and 5-ethyliden-2-norbornene were carried out at room temperature for 1.00 min, using a catalyst/monomer ratio of 1/3000. The results are summarized in Table 2 (runs 1–6).

**Table 2.** ROMP of 2-norbornene and 5-ethyliden-2-norbornene.

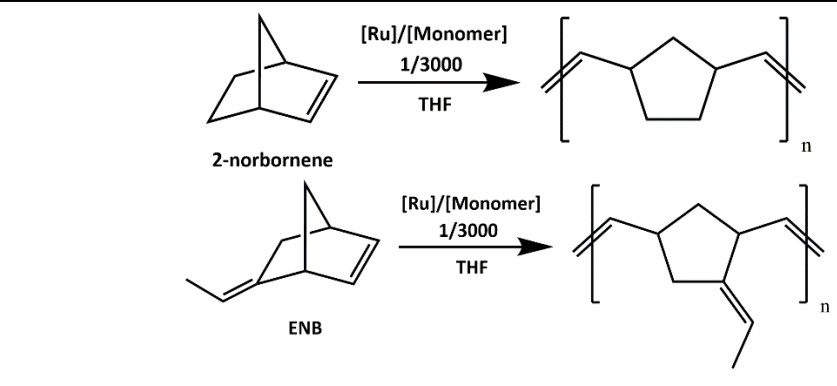

| ᵃ Run | Catalyst | Monomer | Amount of Polymer (g) | Conversion % |
|---|---|---|---|---|
| 1 | HGII | 2-norbornene | 0.991 | 95.3 |
| 2 | *syn*-HGII$_{Ph-Mes}$ | 2-norbornene | 0.908 | 87.4 |
| 3 | *anti*-HGII$_{Ph-Mes}$ | 2-norbornene | 0.997 | 95.9 |
| 4 | HGII | ENB | 1.21 | 91.7 |
| 5 | *syn*-HGII$_{Ph-Mes}$ | ENB | 1.08 | 81.2 |
| 6 | *anti*-HGII$_{Ph-Mes}$ | ENB | 0.670 | 50.4 |
| ᵇ 7 | HGII | ENB | 0.736 | 55.4 |
| ᵇ 8 | *syn*-HII$_{Ph-Mes}$ | ENB | 1.01 | 76.0 |
| ᵇ 9 | *anti*-HGII$_{Ph-Mes}$ | ENB | 0.585 | 44.0 |
| ᶜ 10 | HGII | ENB | 1.31 | 98.7 |
| ᶜ 11 | *syn*-HGII$_{Ph-Mes}$ | ENB | 1.08 | 81.7 |
| ᶜ 12 | *anti*-HGII$_{Ph-Mes}$ | ENB | 1.32 | >99 |

ᵃ Mol cat = $3.72 \times 10^{-6}$; mol monomer $1.11 \times 10^{-2}$; 99 mL THF, 25 °C, 1 min. ᵇ The catalysts were kept at 180 °C for 2 h. ᶜ Run performed at −50 °C for 2 h.

In contrast to *syn*-HGII$_{Ph-Mes}$, which converts to polymer at a slightly lower rate, complexes HGII and *anti*-HGII$_{Ph-Mes}$ provide a conversion of greater than 95% after 60 s. The difference could depend on the necessary activation time for this complex. Results of runs 4–6 (Table 2) of ROMP of 5-ethyliden-2-norbornene show that, in the polymerization of this monomer, the activities of the ruthenium-based catalysts seem apparently lower than those of the ROMP of 2-norbornene, but in this case we obtain a highly cross-linked polymer, therefore the access to the catalytic site by the monomer could be prevented. For instance, *anti*-HGII$_{Ph-Mes}$ produces a polymer with a grade of cross-link higher than 80%.

In order to compare the thermal stability of three ruthenium-based catalysts, they were kept at 180 °C for two hours, after which they were used to catalyze the ROMP of 5-ethyliden-2-norbornene at room temperature (see runs 7–9). Comparing the results of runs 4–6 with those of runs 7–9, it is possible to note that the activities of *syn*- and

*anti*-HGII$_{Ph-Mes}$ are substantially similar, whereas the one of HGII is significantly lower. The results of these tests demonstrate that the two complexes with phenyls on the backbone are much more thermally stable than HGII, so much so that their catalytic activity does not change much, while that of HGII is reduced by about 40%. The complexes also underwent stability tests using thermogravimetry (TGA) and differential scanning calorimetry (DSC). The TGA investigation showed that the complexes are stable up to 230 °C, after which time a degradation process starts with a commensurate weight loss. DSC analyses supported this conclusion as well (see SI). It is also worth noting that HGII is active in the forbidding environment of epoxy resin-based thermosetting material, but only if it is cured at temperatures up to 90 °C.

The catalysts are very efficient at very low temperatures as well. In fact, polymerizations of 5-ethyliden-2-norbornene performed at −50 °C for two hours with all three ruthenium complexes gave very high conversions (see runs 10–12).

## 3. Experimental Part

All reactions were performed in an oxygen- and moisture-free atmosphere using standard Schlenk and glovebox techniques. All solvents were thoroughly deoxygenated and dehydrated under nitrogen atmospheres by heating at reflux over suitable drying agents. NMR-deuterated solvents (Euriso-Top products) were kept in the dark over molecular sieves. Reagents were purchased from Sigma-Aldrich and TCI Chemicals and used without further purification; HGII complex was purchased by Sigma-Aldrich and purified by silica gel chromatography before use. NMR spectra were recorded at room temperature on a Bruker AM 300 spectrometer (300 MHz for $^1$H) and a Bruker AVANCE 400 spectrometer (100 MHz for $^{13}$C). NMR samples were prepared by dissolving about 10 mg of the complex in 0.5 mL of deuterated solvent. The $^1$H NMR and $^{13}$C NMR chemical shifts are referenced to the residual proton impurities of the deuterated solvents with respect to SiMe$_4$ ($\delta$ = 0 ppm) as internal standard singlet signals (CD$_2$Cl$_2$: $\delta_H$ = 5.32, $^{13}$C $\delta_C$ = 53.84). Multiplicities are abbreviated as singlet (s); septuplet (septet); and multiplet (m). Elemental analyses for C, H, and N were obtained by a Thermo-Finnigan Flash EA 1112 according to standard microanalytical procedures. ESI-MS spectra were obtained by using a Waters Quattro Micro triple quadrupole mass spectrometer equipped with an electrospray ion source.

(*R,R*)-1,3-bis(2-mesityl)-4,5-diphenylethylenediamine (A) and imidazolium tetrafluoroborate salt (B) were synthesized following an adapted method previously reported in the literature [49]. Synthesis of *syn*-HGII$_{Ph-Mes}$ was performed following the same procedure described in the literature by some of us [43].

### 3.1. Synthesis of 4R,5R-(1,3-Bis-mesityl)-(4,5-diphenyl-imidazolin-2-ylidene)-dichloro-(2-isopropoxybenzylidene)-ruthenium(II) (Anti-HGII$_{Ph-Mes}$)

In a glovebox, a 50-mL flask was charged with imidazolinium salt (B) (0.340 g, 0.620 mmol) and potassium hexamethyldisilazide (0.136 g, 0.682 mmol) in dry toluene. After a few minutes, the first-generation Hoveyda–Grubbs catalyst (0.196 g, 0.329 mmol) was added. The reaction mixture was heated to 70 °C for four hours. After this time, the reaction mixture was concentrated and purified by flash column silica gel chromatography (hexane/diethyl ether from 5/1 to 3/1) to afford the desired ruthenium catalyst as a green solid (7.72 × 10$^{-2}$ g, yield 30%).

$^1$H NMR (300 MHz, CD$_2$Cl$_2$): $\delta$ 16.3$_5$ (s, 1H, Ru = C*H*Ph); 7.5$_7$–6.8$_4$ (m, 18H, aromatic carbons); 5.8$_6$ (s, 2H, N(C*H*Ph)$_2$N); 4.8$_7$ (septet, 1H, (CH$_3$)$_2$C*H*OAr); 2.8$_8$–1.1$_8$ (24H, OCH(C*H$_3$*)$_2$ + 6xC*H$_3$* mesityl). $^{13}$C{$^1$H} NMR (100 MHz, CD$_2$Cl$_2$): $\delta$ 298.3 (Ru = CH–*o*OiPrC$_6$H$_4$); 214.5 (NCN), 152.5; 145.7; 141.3; 140.0; 139.4; 138.1; 137.2; 135.6; 133.2; 130.6; 130.2; 130.1; 129.9; 129.7; 129.0; 122.8; 113.4; 75.6 (O*C*(CH$_3$)$_2$); 74.5 (N(*C*HPh)$_2$N); 73.4 (N(*C*HPh)$_2$N); 32.1; 23.2; 22.2; 21.3; 19.7; 19.4 (methyl groups). Anal. Calcd (%) for C$_{43}$H$_{46}$Cl$_2$N$_2$ORu (778.20): C 66.31, H 5.95, N 3.60. Found: C 66.22, H 6.02, N 3.52. ESI-MS, (m/z), 742.9 Dalton [C$_{43}$H$_{46}$ClN$_2$ORu]$^+$, 783.8 Dalton [C$_{43}$H$_{47}$ClN$_2$ORuK]$^+$.

*3.2. Ring Closing Metathesis*

3.2.1. RCM of Diethyl-diallylmalonate (C)

An NMR tube with a screw-cap septum top was charged inside a glovebox with 0.80 mL of a solution of the catalyst (0.1%) in $C_6D_6$. The NMR tube was equilibrated at 60 °C in the NMR probe before **C** (19.3 µL, 19.2 mg, 0.080 mmol, 0.1 M) was added via syringe. The conversion to **D** was determined by integrating the methylene protons of the reagent at δ $2.8_4$ (dt) and of the product at δ $3.1_4$ (s).

3.2.2. RCM of Diethyl-allylmethallylmalonate (E)

An NMR tube with a screw-cap septum top was charged inside a glovebox with 0.80 mL of a solution of the catalyst (0.1%) in $C_6D_6$. A 20.5 µL portion of E was injected into a heated NMR tube containing 0.80 mL of catalyst solution (0.1 mol %). The conversion to F was determined by integrating the methylene protons of the reagent at δ $2.9_6$ (d), $2.9_3$ (s), and of the product at δ $3.1_8$ (m), $3.0_7$ (s).

3.2.3. RCM of Diethyl-dimethallylmalonate (G)

An NMR tube with a screw-cap septum top was charged inside a glovebox with catalyst stock solution (0.016 M, 250 µL, 4.0 µmol, 5.0 mol%) and $C_6D_6$ (0.55 mL). Olefin G (21.6 µL, 21.5 mg, 0.080 mmol, 0.1 M) was added via syringe, and the sample was placed in an oil bath regulated at 60 °C. The conversion to H was determined by comparing the ratios of the integrals of the methylene protons in the starting material, δ $2.9_7$ (s), and of the product at δ $3.1_4$ (s) for the different times.

*3.3. Ring Opening Metathesis Polymerization*

3.3.1. ROMP of 1,5-Cyclooctadiene (COD)

An NMR tube with a screw-cap septum top was charged with 0.80 mL of a $C_6D_6$ solution of catalyst (0.40 µmol). After equilibrating at 30 °C the sample in the NMR probe, 49.1 µL (0.40 mmol) of COD was injected into the tube. The polymerization was monitored as a function of time, and the conversion to poly-COD was determined by integrating the methylene protons in the starting monomer, δ $2.3_0$ (m), and those in the product, δ $2.1_3$ (br m), and $2.1_1$ (br m).

3.3.2. ROMP of 2-Norbornene and 5-Ethyliden-2-Norbornene

Polymerization of 2-norbornene (or 5-ethyliden-2-norbornene) was carried out in a flask equipped with a magnetic stirrer at ambient temperature. In a typical experiment, a solution of initiator ($3.72 \times 10^{-6}$ mol in 1 mL of THF) was mixed with a stirred solution of $1.11 \times 10^{-2}$ mol of monomer in 99 mL of THF. The reaction was terminated by adding a few drops of ethyl vinyl ether. The polymers were precipitated in methanol, recovered by filtration, washed with hexane, and dried in a vacuum.

**4. Conclusions**

Since the development of Hoveyda-Grubbs 2nd generation of catalysts in the early 2000s, countless modifications of the imidazolium nitrogen substituents as well as the *ortho*-isopropoxybenzylidene moiety (often referred to as the Hoveyda chelate) have been reported. At the same time, fewer examples of imidazolium's backbone structural changes were documented. In this paper, novel Hoveyda-Grubbs type II catalysts functionalized with two phenyl substituents on the imidazolium ring, either in *syn* or *anti* configuration, are reported.

These, compared with the HGII classical complex, display similar, high efficiency in the RCM catalysis of olefins also sterically hindered. The same catalysts are very active in ROMP even at extremely low temperatures (−50 °C). At room temperature, the *anti*-HGII$_{Ph-Mes}$ complex produces a highly cross-linked polymer with 5-ethylidene-2-norbornene; hence, this may have an impact on its activity, making it less active than the HGII and *syn*-HGII$_{Ph-Mes}$ complexes. The most remarkable result is that HGII$_{Ph-Mes}$

complexes are significantly thermally stable since their activity in the ROMP of 5-ethylidene-2-norbornene does not change by conducting the polymerization at room temperature after holding the complexes for two hours at 180 °C. This does not account for HGII, making these derivatives especially interesting for material applications such as self-healing composite materials.

**Supplementary Materials:** The following supporting information can be downloaded at: https://www.mdpi.com/article/10.3390/inorganics11060244/s1, Figure S1: $^1$H NMR (300 MHz, $CD_2Cl_2$): $\delta$ 16.3$_5$ (s, 1H, Ru = C*H*Ph); 7.5$_7$–6.8$_4$ (m, 18H, aromatic carbons); 5.8$_6$ (s, 2H, N(C*H*Ph)$_2$N); 4.8$_7$ (septet, 1H, (CH$_3$)$_2$C*H*OAr); 2.8$_8$–1.1$_8$ (24H, OCH(C*H$_3$*)$_2$ + 6xC*H$_3$* mesityl); Figure S2: $^{13}$C{$^1$H} NMR (100 MHz, $CD_2Cl_2$): $\delta$ 298.3 (Ru = C*H*–*o*OiPrC$_6$H$_4$); 214.5 (NCN), 152.5; 145.7; 141.3; 140.0; 139.4; 138.1; 137.2; 135.6; 133.2; 130.6; 130.2; 130.1; 129.9; 129.7; 129.0; 122.8; 113.4; 75.6 (O*C*(CH$_3$)$_2$); 74.5 (N(*C*HPh)$_2$N); 73.4 (N(*C*HPh)$_2$N); 32.1; 23.2; 22.2; 21.3; 19.7; 19.4 (methyl groups); Figure S3: Thermogravimetric analysis (TGA) of HGII; Figure S4: Thermogravimetric analysis (TGA) of HGII$_{Ph-Mes}$; Figure S5: Differential scan calorimetry analysis (DSC) of HGII; Figure S6: Differential scan calorimetry analysis (DSC) of HGII$_{Ph-Mes}$.

**Author Contributions:** Conceptualization, P.L., A.M. and A.D.; methodology, A.M. and A.D.; data curation, A.M. and A.D.; writing—original draft preparation, P.L., A.M. and A.D.; supervision, P.L. All authors have read and agreed to the published version of the manuscript.

**Funding:** This research received no external funding.

**Institutional Review Board Statement:** Not applicable.

**Informed Consent Statement:** Not applicable.

**Data Availability Statement:** The data presented in this study are available in this article and Supplementary Materials.

**Acknowledgments:** The authors are grateful to Patrizia Oliva and Patrizia Iannece for the technical assistance.

**Conflicts of Interest:** The authors declare no conflict of interest.

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
