# Peer review of "New Insights into the Catalytic Activity of Second Generation Hoveyda–Grubbs Complexes Having Phenyl Substituents on the Backbone"

_inorganics, doi:10.3390/inorganics11060244_

Round 1

Reviewer 1 Report

This manuscript reports the reactivities of two Hoveyda-Grubbs catalyst derivatives with the modification of the NHC backbone with two phenyl groups, where the phenyl groups are arranged in syn- and anti-configurations.

The introduction section states that this work was arranged based on the authors’ previous results and the difference in the thermal reactivities of Hoveyda-Grubbs catalyst derivatives with a syn- or anti-diphenyl NHC ligand, meaning that the motivation of this work is the difference in the modified Hoveyda-Grubbs catalysts with syn- and anti-diphenyl configurations. However, most parts describe the comparison of the reactivities between HG-II and the modified HG-II derivatives (especially, RCM reactions). If the authors intend to say that no reactivity difference between the modified HG-II derivatives for RCM reactions, the description should follow as that. The entire manuscript requires rewriting.

The reviewer must point out the following more critical points: (1) Characterization of compounds:

Both catalysts with syn- and anti-configurations are known compounds (i.e., previously reported compounds). Therefore, the authors should indicate that the spectroscopic data for the synthesized catalysts follow those previously reported. How did the authors confirm the stereoconfigurations of the synthesized catalysts? This is the central point of catalyst characterization in this study. Unless the confirmation of stereo-configurations of the synthesized catalysts is clear, this work is not acceptable. (2) Experimental procedure:

The RCM reactions and ROMP of COD were monitored using NMR, where yields are determined every 1 min. As desribed in the experimental section, reaction samples were prepared in a glovebox and heated in the oil both. In this procedure, NMR measurements  in short time periods after the reaction initiation appears to be impractical.  Considering the accumulation time in NMR measurements, acquisition of NMR data is unlikely. Moreover, no error bars is indicated the time profiles; the bias in the yield determination should be considered in NMR monitoring (generally, ca 5%). This means that the description in the RCM of compound G (“in fact, HGII achieves a conversion of 63%, while anti-HGIIPh-Mes and syn-HGIIPh-Mes reach about 60 and 55%, respectively. As a result, whereas syn-HGIIPh-Mes and anti-HGIIPh-Mes exhibit lower initial activity than HGII, over a long period of time, they behave similarly.”) should revised. In the final phase of the reactions, the yields are similar between the three catalysts. 

(3) discussion on the ROMP of COD

The authors proposed that the steric effect is the origin of the induction period observed in the syn-configuration catalyst-mediated ROMP of COD. However, the phenyl groups are located far from the coordination site of a substrate molecule (the mesityl moiety is a more critical factor for the substrate access). To confirm and insist on the steric effect, the authors must discuss the occurrence of steric effect indicating X-ray crystallographic structures for the two modified HG-II catalysts.

Author Response

The authors thank Reviewer 1 for the useful comments.

In this work, the synthesis and the characterization of the 4R,5R-(1,3-bis-mesityl)-(4,5-diphenyl-imidazolin-2-ylidene)-dichloro-(2-isopropoxybenzylidene)-ruthenium(II) (anti-HGIIPh-Mes) complex was reported. The activity of this complex was compared with the one of the analogous compound having the phenyls in positions 4 and 5 of the ligand in syn configuration (syn-HGIIPh-Mes). The effect of these backbone substituents on the catalytic activity of the related complexes was determined by comparison with the commercial Hoveyda-Grubbs’ catalyst (1,3-bis-mesityl)-(imidazolin-2-ylidene)-dichloro-(2-isopropoxybenzylidene)-ruthenium(II) (HGII) in RCM and ROMP. All the catalysts are highly efficient in the RCM catalysis of olefins, also sterically hindered, and in ROMP even at extremely low temperatures (-50 °C). The result obtained certainly deserves to be underlined as it relates to the thermal stability of the complexes having phenyl groups on the backbone of the ligand, in fact their activities are substantially very similar by conducting the ROMP of 5-ethylidene-2-norbornene at room temperature and after holding the complexes for two hours at 180 °C, while this does not account for HGII.

  • The spectroscopic data of the syn-HGIIPhe-Mes complex are perfectly consistent with those reported in the literature. (Ref [42] Longo, P.; Mariconda, A.; Calabrese, E.; Raimondo, M.; Naddeo, C.; Vertuccio, L.; Russo, S.; Iannuzzo, G.; Guadagno, L. Development of a New Stable Ruthenium Initiator Suitably Designed for Self-Repairing Applications in High Reactive Environments. Ind. Eng. Chem. 2017, 54, 234–251, doi:10.1016/j.jiec.2017.05.038).

The absolute configuration of carbons 4 and 5 of the ligands is exactly the same as the starting amines, it is not modified during the synthesis of the complexes. This has been verified in previous works also by some of us as well, i.e.: Organometallics 2008, 27, 4649–4656; [Ref. 37] Organometallics 2009, 28, 4988–4995; [Ref. 38] Chem. Eur. J. 2011, 17, 8618 – 8629; [Ref. 48] Organometallics 2017, 36, 3692−3708.

  • Kinetic profile of RCM reactions and ROMP of COD monitored using NMR.

The RCM reactions of diallylmalonate and diethyl allylmethallylmalonate were performed in an NMR tube with a screw-cap septum top. It was charged inside a glovebox with 0.80 mL of a solution of the catalyst (0.1%) in C6D6. The NMR tube was equilibrated at 60 °C in the NMR probe before that the suitable diolefin was added via syringe. The conversions to cyclopentene derivatives were determined by integrating the methylene protons of the reagents and the ones of the products at different times.

The RCM reaction of diethyl dimethylallylmalonate was performed in an NMR tube with a screw-cap septum top was charged inside a glovebox with catalyst stock solution and C6D6. Diethyl dimethylallylmalonate was added via syringe, and the sample placed in an oil bath regulated at 60 °C. The conversion to H was determined by comparing the ratio of the integrals of the methylene protons in the starting material and of the product for the different times (1h, 24h, 48h, 72h and 96h).

ROMP of 1,5-cyclooctadiene (COD) was carried out in an NMR tube with a screw-cap septum top charged with 0.80 mL of a C6D6 solution of the catalyst (0.40 μmol). After equilibrating at 30 °C the sample in the NMR probe, 49.1 μL (0.40 mmol) of COD was injected into the tube. The polymerization was monitored as a function of time, and the conversion to poly-COD was determined by integrating the methylene protons in the starting monomer and those in the product.

It is worth noting that all tests were performed in triplicate, resulting in excellent reproducibility. In fact, performing the 1H NMR experiments with a one minute relaxation delay yielded differences of less than 1%.

  • As described by Cavallo and Costabile (J. Am. Chem. Soc. 2004, 126, 9592-9600) the two Ph groups in positions 4 and 5 of the imidazolyl ring of the NHC ligand impose a folding to N-bonded aromatic groups. In particular, the N-bonded aromatic group near the Ph group is bent down, that is bent toward the Ru atom and the substrate. This structural feature is also found in the crystal structure of the (pre)catalyst. (Seiders, T. J.; Ward, D. W.; Grubbs, R. H. Org. Lett. 2001, 3, 3225).

Reviewer 2 Report

The paper reports the synthesis of slightly modified Grubbs-Howeida catalysts on which two phenyl rings have been added, syn and anti to each other, on the carbene unit of the catalyst. This modification leads to catalysts whose properties were studied and found to have similar performances as the original catalysts. However the modification of the catalysts provide some significantly improved thermal stability to the HG catalysts, this being an additional quality to the known catalysts that may be useful for specific application of the catalysts when used for the preparation of material devices. I do not have important improvements to suggest for the paper and thus I recommend its publication afetr slight modifications in Inorganics.

The english does apparently not require extensive improvements.

Author Response

The authors thank Reviewer 2 for the kind coomment. Please note that the manuscript has been improved in some parts.

Reviewer 3 Report

In this manuscript, the authors reported the synthesis of Hoveyda-Grubbs’ type II complex bearing two anti-phenyl backbone substituents with mesityl N-substituents. The comparison of the catalytic performances of the complex with its syn-configuration complex and classic Hoveyda-Grubbs II complex was reported. The kinetic during the ring closing metathesis of a series of compounds, as well as the ring opening metathesis polymerization of a series of compounds, were studied. There is no doubt that the work is valuable, however, some major problems should be addressed before the acceptance of the manuscript.
1. Most of the references are beyond five years, is there any recent report that can be cited?
2. The thermal stability of the reported anti-HGII Ph-Mes should be added.
3. The authors should further state the innovation of this research, a similar research can be found in Organometallics 2017, 36, 3692−3708. The authors should compare their results with this report and cite this paper in the manuscript.

Author Response

The authors express their gratitude to the reviewer for her/his positive comments.

  • As suggested by the reviewer some more recent references have been added:
  1. Catalysts (2023), 13, 34 [Ref. 34]
  2. Soc. Rev. (2018), 47, 4510 [Ref. 30]
  3. Molecules (2016), 21, 117 [Ref. 31]
  4. Rec. (2021), 21, 3648 [Ref. 35]
  5. Organometallics (2017), 36, 3692 [Ref. 48]
  • The activity of the complexes in the 5-ethyliden-2-norbornene ROMP, after being kept at 180 °C for two hours, is certainly a clear indication of the thermal stability of the tested complexes. Furthermore, thermogravimetric and differential scanning calorimetry analyses were performed on the complexes. By means of the TGA analysis it resulted that the complexes are stable up to 230°C, then a degradation begins with a corresponding loss in weight. This finding was confirmed by DSC analyses for both syn and anti-HGIIPh-Mes, in fact, an exothermic peak is showed in the thermogram of these compounds. Instead, DSC thermogram of HGII show an  endothermic peak around 210°C, followed by exothermic peak at 230°C. Probably, the endothermic peak is due to a possible structural modification that is not easily identifiable.    The data of such analyses have been described in the section “results and discussion” of the manuscript, while the thermograms of HGII and anti-HGIIPh-Mes (as reference for both syn and anti-HGIIPh-Mes, being identical) have been reported in the SI.

  • In the paper Organometallics 2017, 36, 3692−3708 is reported that the NHC substitution patterns strongly influence the stability and reactivity of corresponding complexes. In fact, complexes bearing an anti NHC backbone are more stable and more active than their corresponding syn isomers. In the cited paper the effects of different substituents to the nitrogen atoms with respect to the syn or anti configuration of the phenyls on the back-bone of the carbenic ligand are compared. In the present work the substituents of the nitrogen atoms are the same, therefore only the effect of the different configuration of the phenyls in position 4 and 5 of the ring can be evaluated. Our catalysts are highly efficient both in the RCM catalysis of olefin and in ROMP. Moreover, the complexes (both syn and anti-HGIIPh-Mes) have a high thermal stability because their activities are substantially very similar by conducting the ROMP of 5-ethylidene-2-norbornene at room temperature and after holding the complexes for two hours at 180 °C, while this does not account for commercial HGII. The results of the paper (Organometallics 2017, 36, 3692−3708) have been opportunely cited in the manuscript.

This sentence was inserted in the manuscript (line 194):

It may be useful to recall that, as reported by Grisi et al. (Organometallics 2017, 36, 3692−3708), the stability and reactivity of complexes with different substituents on the nitrogen atoms of NHC are strongly influenced by the syn or anti configuration of the phenyl groups on the ligand backbone. In fact, the complex with the phenyls in anti configuration is more stable and active than the homologue in syn configuration. In the present work only the stereochemical effects of the phenyls in position 4 and 5 are evaluated, in the presence of the same mesityl substituents on the nitrogen atoms. Our catalysts are highly efficient in both RCM catalysis of olefins and ROMP.

Reviewer 4 Report

This manuscript offers a follow-up study from a 2017 report on phenyl substituents introduced to the NHC ligand of Hoveyda-Grubbs’ like metathesis catalysts.  In the examination of various metathesis procedures, the authors note that the phenyl substituents generally reduce metathesis activity slightly but provide great thermal stability to the catalysts.  As reported earlier, this could provide some advantages in the development of thermally initiated self-healing polymers.  While the contribution is incremental in nature, there are some that might be interested in this work.  Before publication, however, there are several issues (some significant) that need to be addressed.  

Abstract, lines 11+12:  The following first two sentences of the introduction needs to be clarified.   “…is olefin metathesis.  This [2+2] cycloaddition is…”  Olefin metathesis is not a cycloaddition.  While the mechanism of the reaction certainly involves the formation and disappearance of metalocyclobutanes via [2+2] cycloadditions and retrocycloaddition, the olefin metathesis reaction in not a cycloaddition and shouldn’t be charactered as such.

Line 38:  Grubbs catalysts should be Grubbs’ catalysts.

Figure 4: The graph has no valuable kinetic information after approx. 30 minutes.  The graph should be redrawn where the more telling 0-30 mins points are shown more clearly.  

Lines 177-181: This key speculation regarding their compound stability could benefit with a figure illustrating the author’s point.

Lines 227-228:  The authors indicate in their experimental section that their silica gel purified precatalysts were compared with commercially available HGII (usually collected by crystallization).  Unfortunately, it has been our experience that commercial sources of HGII can come with a broad spectrum of impurities that can effect the stability and side reactions typically associated with olefin metathesis reactions.  To get reliable and reproducible kinetic, stability and other studies, it is imperative that the commercial HGII is also purified by silica gel chromatography before comparing with the author’s purified catalysts.  This was not indicated, which sheds a major concern regarding the value of author findings.  

Publication is only warranted after addressing these major concerns.

see comments for authors section

Author Response

The authors express their gratitude to the reviewer for her/his positive comments.

Abstract, lines 11+12:  

As suggested by the referee, we have changed the two sentences in the abstract and introduction:

  • These useful reactions are commonly promoted by ruthenium-based precatalysts, namely the second generation Grubbs’ catalyst (GII) and complexes bearing a styrenyl ether ligand, referred to as the second generation Hoveyda-Grubbs’ catalyst (HGII).
  • The accepted reaction mechanism, characterized by the formation of a metallacyclobutane intermediate, was first proposed by Chauvin [27].

Line 38:  Grubbs catalysts should be Grubbs’ catalysts.

The suggested change has been made.

Figure 4: The graph has no valuable kinetic information after approx. 30 minutes. The graph should be redrawn where the more telling 0-30 mins points are shown more clearly.  

The suggested change has been made.

Lines 177-181: This key speculation regarding their compound stability could benefit with a figure illustrating the author’s point.

The activity of the complexes in the 5-ethyliden-2-norbornene ROMP, after being kept at 180 °C for two hours,  is certainly a clear indication of the thermal stability of the tested complexes. Furthermore, thermogravimetric and differential scanning calorimetry analyses were performed on the complexes. By means of the TGA analysis it resulted that the complexes are stable up to 230°C, then a degradation begins with a corresponding loss in weight. This finding was confirmed by DSC analyses for both syn and anti-HGIIPh-Mes, in fact, an exothermic peak is showed in the thermogram of these compounds. Instead, DSC thermogram of HGII show an  endothermic peak around 210°C, followed by exothermic peak at 230°C. Probably, the endothermic peak is due to a possible structural modification that is not easily identifiable.    The data of such analyses have been described in the section “results and discussion” of the manuscript, while the thermograms of HGII and anti-HGIIPh-Mes (as reference for both syn and anti-HGIIPh-Mes, being identical) have been reported in the SI.

Lines 227-228:  The authors indicate in their experimental section that their silica gel purified precatalysts were compared with commercially available HGII (usually collected by crystallization).  Unfortunately, it has been our experience that commercial sources of HGII can come with a broad spectrum of impurities that can effect the stability and side reactions typically associated with olefin metathesis reactions.  To get reliable and reproducible kinetic, stability and other studies, it is imperative that the commercial HGII is also purified by silica gel chromatography before comparing with the author’s purified catalysts.  This was not indicated, which sheds a major concern regarding the value of author findings.  

The authors thank the reviewer. HGII was effectively purified by silica gel chromatography before use. This has been indicated in the experimental part.

Round 2

Reviewer 1 Report

This reviewer found that the authors revised some points in the manuscript. However, they still did not fully answer my critism regarding the NMR measurements. According to the authors' response, the sample tube was incubated in the NMR probe (in the first manuscript draft, the authors described the incubation of the sample in an oil bath, not in the NMR probe). I can understand it.  For NMR measurements, we should perform several procedures before data accumulation (signal lock, shim adjustment, etc), which takes several minutes. And then, the acuquired data in the initial phase of the reactions (shown in Figures 4 and 5) are not reliable. The data in the initial phase of the reactions should be removed.

Author Response

Comments of Reviewer 1:

This reviewer found that the authors revised some points in the manuscript. However, they still did not fully answer my criticism regarding the NMR measurements. According to the authors' response, the sample tube was incubated in the NMR probe (in the first manuscript draft, the authors described the incubation of the sample in an oil bath, not in the NMR probe). I can understand it.  For NMR measurements, we should perform several procedures before data accumulation (signal lock, shim adjustment, etc), which takes several minutes. And then, the acquired data in the initial phase of the reactions (shown in Figures 4 and 5) are not reliable. The data in the initial phase of the reactions should be removed.

Answers to Reviewer 1:

The authors point out that even in the first draft the procedures for obtaining the kinetic data of the RCMs were described in a precise way, probably in a somewhat cryptical manner, so it may be appropriate to explain them better:

  • For diolefins diallylmalonate and diethyl allylmethallylmalonate the procedure requires that a solution of 0.80 mL of C6D6 with 0.1% of the catalyst is prepared in a glove-box in an NMR tube with a screw-cap septum top. Then, the solution is equilibrated in the NMR spectrometer probe at 60 °C, performing the operations relative to signal lock, shim adjustment, rga etc.

The NMR tube has been removed from the instrument and the olefin has been quickly added via syringe. At the same time as the addition of the olefin, the time taken for the instrument to record the first NMR spectrum and then subsequent ones for each minute was recorded. The first experiment starts on average after 1 minute and thirty seconds.

The sequence of these operations is such that the kinetic data reported are certainly reliable.

 The kinetic profile, for the first points of diallylmalonate, is:

HG2

syn HG2Ph-Mes

anti HG2Ph-Mes

Time

(min)

Conversion (%)

Time

(min)

Conversion

(%)

Time

(min)

Conversion

(%)

1,47

87,09

1,09

23,98

1,25

23,2

2,47

97,59

2,09

66,69

2,25

74,4

3,47

99,69

3,09

87,25

3,25

94,3

…..

…..

….

 It is worth noting that this same procedure has been reported in many papers in the literature, e.g.:

T. Ritter, A. Hejl, A.G. Wenzel, T.W. Funk, and R.H. Grubbs A Standard System of Characterization for Olefin Metathesis Catalysts Organometallics 2006, 25, 5740-5745;        

 - see also Ref [33-36].

  • The incubation in the oil bath at 60 °C is done only for the tetrasubstituted diolefin (diethyl-allyl-dimethyl-malonate), in fact in this case the first kinetic data is taken after one hour, 24, 48, 72 and 96 hours.

Reviewer 3 Report

The authors have made the necessary revisions, thus I suggest the acceptance of the manuscript.

Author Response

The authors thank the reviewer for her/his positive comment.

Reviewer 4 Report

The authors have addressed effectively the initial issues with the manuscript.  Publication is now suggested.

Author Response

(The authors gave the same response as above.)
